# Probiotic Fermentation of *Astragalus membranaceus* and Raphani Semen Ameliorates Cyclophosphamide-Induced Immunosuppression Through Intestinal Short-Chain Fatty Acid-Dependent or -Independent Regulation of B Cell Function

**DOI:** 10.3390/biology14030312

**Published:** 2025-03-19

**Authors:** Yang Chen, Xiaoqing Wei, Binqi Rui, Yutong Du, Zengjie Lei, Xiujie Guo, Chaoran Wang, Donglin Yuan, Xiuli Wang, Ming Li, Binbin Hou, Yinhui Liu

**Affiliations:** 1College of Basic Medical Science, Dalian Medical University, Dalian 116044, China; yangchen@dmu.edu.cn (Y.C.); jcyxyyb_dy@163.com (X.W.); ruibinqi7270@163.com (B.R.); ddyttyqsl@163.com (Y.D.); 15382193852@163.com (Z.L.); panpan1210_dmu@163.com (X.W.); vivianmarat@163.com (M.L.); 2CAS Key Laboratory of Separation Science for Analytical Chemistry, Dalian Institute of Chemical Physics, Chinese Academy of Sciences, Dalian 116023, China; guoxiujie@castim.cn (X.G.); wcrpj_505@dicp.ac.cn (C.W.); 3Department of Dermatology, The Second Affiliated Hospital of Dalian Medical University, Dalian 116023, China; donghandmd@163.com

**Keywords:** *Astragalus membranaceus*, Raphani Semen, probiotic fermentation, immunity, gut microbiota, SCFA

## Abstract

Traditional Chinese medicine offers significant benefits to human health, yet its application faces considerable limitations. The development and utilization of Chinese herbs can be enhanced through probiotic fermentation. In this study, three strains of probiotics, *Bifidobacterium longum* SD5219, *Lactobacillus fermentum* NCIMB5221, and *Lactobacillus para-casei* SD5219, were used to ferment *Astragalus membranaceus* and Raphani Semen, investigating the effects of these fermentation products on the immune function of immunosuppressed mice. The results demonstrated that probiotic fermentation could increase the conversion rate of active compounds in traditional Chinese medicine and promote probiotic proliferation. This process altered the gut microbiota structure in mice, increasing the abundance of beneficial bacteria and promoting short-chain fatty acid production. It also repaired the intestinal mucosal barrier, enhanced humoral immunity, and effectively restored the immunosuppressive state in mice. In summary, probiotic fermentation of *Astragalus membranaceus* and Raphani Semen can enhance the transformation of active components in traditional Chinese medicine, restore intestinal microecological health, and improve immunosuppression in mice. These findings provide a foundation for the clinical application of fermented traditional Chinese medicine as an immunomodulatory agent.

## 1. Introduction

The human immune system is a complex and evolving network that protects the body against a variety of diseases, including infectious, inflammatory, autoimmune, neoplastic, and neurodegenerative diseases [1,2]. The human gastrointestinal tract is the most important digestive organ, as well as an immune organ that is colonized by a large number of microorganisms. The gut microbiome co-evolves with the host and plays an important role in maintaining intestinal homeostasis and immune regulation [3,4]. The gut microbiota produces numerous beneficial substances for the host, such as short-chain fatty acids (SCFA), bile acids, tryptophan sphingolipids, and vitamins [5]. Alterations in influencing factors, such as host genotypes, physiological status, diet, drugs, and living conditions, lead to disturbances in the gut microbiota and cause diseases, including intestinal barrier dysfunction, low-grade chronic systemic inflammation, and a variety of immune-related disorders [6,7]. The compromised immune system, in turn, affects the intestinal mucosal barrier and causes gut microbiota disorders [5,8]. Therefore, a close relationship exists between immune function and intestinal barrier integrity. Improving human intestinal immunity has become a popular research topic in recent years [9].

Extensive practical research has been conducted on the modulation of immunity and repair of intestinal barrier dysfunction in traditional Chinese medicine (TCM) [10,11]. The valuable Chinese herbal medicine *Astragalus membranaceus* (Fisch.) Bunge (*A. membranaceus*) has been used in China for over 2000 years and has been documented in Shennong’s Classic of Materia Medica [12]. The major components of *A. membranaceus* include polysaccharides, saponins, and flavonoids. The contemporary use of *A. membranaceus* primarily focuses on its immunomodulatory, antioxidant, anti-inflammatory, and anticancer properties [12]. Astragalus polysaccharides enhance the activities of diverse immune cells and induce the expression of various cytokines [13]. Raphani Semen (Lai Fu-zi in Chinese), the dried seed of *Raphanus sativus* L., is another traditional Chinese herbal medicine recorded in the *Chinese Pharmacopoeia* (2020). Its chemical composition primarily consists of glucosinolates, sulfur-containing derivatives, and disulforaphenes. Notably, glucosinolates can be enzymatically converted into isothiocyanates by intestinal microbiota or plant myrosinase, which exhibit a wide range of activities, including anti-inflammatory, antitumor, and antioxidant effects [14]. Our previous study demonstrated that dissulforaphenes (isothiocyanates) restored gut microbiota dysbiosis, reversed colitis, and improved intestinal permeability [15].

### Germ

In the clinical application of TCM, water decoctions and pills are predominantly utilized and administered typically through oral ingestion [16]. However, many of these natural substances undergo minimal digestion in the small intestine and are fully broken down through fermentation by a diverse range of gut microorganisms in the colon and cecum [17]. The intestinal microbial community plays a crucial role in the metabolism and bioavailability of specific active components in TCMs [18]; thus, the absorption efficiency of TCMs is enhanced, and the harmonious interaction between natural substances and microorganisms is promoted.

Several previous studies have demonstrated that in vitro microbial fermentation has the potential to standardize TCM products and enhance their clinical efficacy [19]. Fermentation of TCM also positively affects the gut microbiome and host immune system. Our previous research showed that fermentation of *Ganoderma lucidum* and Raphani Semen with probiotic strains, including *Bifidobacterium* and *Lactobacillus* spp., regulated the intestinal mucosal barrier and immune responses in immunosuppressed mice [20]. However, to the best of our knowledge, only a few studies have investigated the immunomodulatory effects of probiotic-fermented *A. membranaceus* and Raphani Semen.

Therefore, the purpose of our study was to evaluate the regulating effects of *A. membranaceus* and Raphani Semen extracts, which were fermented with *B. longum* SD5219, *Lactobacillus fermentum* NCIMB5221, and *Lactobacillus paracasei* SD5219, on immune function, intestinal barrier integrity, and intestinal microorganism in an immunosuppressed mouse model that was induced by CTX as well as the underlying mechanism of prevention and treatment. The present study provides novel insights into enhancing the application of the *A. membranaceus* and Raphani Semen as immunomodulators.

## 2. Materials and Methods

### 2.1. Probiotic Fermentation of A. membranaceus and Raphani Semen

The fine powders of *A. membranaceus* and Raphani Semen were provided by Jiakai Pharmaceutical Co., Ltd. (Hefei, China) and Shanghai Jinliang Food Technology Co., Ltd. (Shanghai, China), respectively. The preparation of fermented *A. membranaceus* and Raphani Semen was conducted following the methodology previously described [21]. In brief, the powders of *A. membranaceus* (100 g) and Raphani Semen (150 g) were separately boiled in 1 L of demineralized water. The water extracts of *A. membranaceus* and Raphani Semen after 0.45 µm polyethersulfone ultrafiltration membrane filtration were added to MRS fluid medium (formulated according to TCM) with concentrations of 150 mg/mL and 500 mg/mL, respectively. Then, the sample was subjected to sterilization at a temperature of 121 °C for 15 min. Subsequently, it underwent filtration through an aseptic 0.22 µm polyethersulfone ultrafiltration membrane that was utilized for obtaining extracts of *A. membranaceus* and Raphani Semen extracts (AS). The sterilized AS was inoculated with 1% (*v*/*v*) activated *B. longum* SD5219, *L. fermentum* NCIMB5221, and *L. paracasei* DM2806, which were maintained in the Culture Collection at Dalian Medical University (Dalian, China) for fermentation under an anaerobic system at 37 °C for 48 h. The fermentation solution was subjected to serial twofold dilution every 12 h. Viable bacterial counts were determined using drop plate counting, and the growth curve was constructed over a 48 h period. This method was employed to assess the growth kinetics of probiotics and determine the optimal fermentation duration.

### 2.2. Component Analysis of the Fermented A. membranaceus and Raphani Semen

In brief, 1 mL of the fermentation broth from *A. membranaceus* and Raphani Semen, which had been fermented with 1% (*v*/*v*) *B. longum* SD5219, *L. fermentum* NCIMB5221, and *L. paracasei* DM2806, was collected and centrifuged at 4 °C for 5 min at 4000 rpm; the supernatant was identified as PROAS. High-performance liquid chromatography–tandem mass spectrometry (HPLC-MS) was used to determine the alterations in the chemical composition of the water extracts before (AS) and after (PROAS) fermentation. The mobile phase was composed of acetonitrile (A) and water with 0.1% formic acid (B), and the linear elution gradient was set referring to the methodology we described previously [22]. The chromatographic separation system comprised a Shimadzu (Kyoto, Japan) LC-30 CE series modular platform, which included a CBM-20A system controller, dual LC-30 CE solvent delivery units, an SIL-30AC automated sampler, an integrated degassing module, and a temperature-controlled column compartment. Analytical separation was performed using a reversed-phase column (2.1 mm × 100 mm, 3 µm particle size) (Waters ACQUITY UPLC HSS T3 C18, Milford, MA, USA) under the following conditions: Mobile phase: binary solvent system consisting of (A) acetonitrile and (B) 0.1% (*v*/*v*) aqueous formic acid. Gradient program (Version 1.2): initiation at 30% A (0–0.5 min), linearly increased to 50% A (0.5–2 min), ramped to 95% A (2–6.5 min), and maintained at 95% A (6.5–8 min). Flow rate: 0.4 mL/min, with an injection volume of 3 µL. Thermo-regulation: Column temperature stabilized at 25 °C. Mass spectrometric detection was conducted using an AB Sciex X500B (SCIEX, Redwood, CA, USA) quadrupole time-of-flight (Q-TOF) instrument equipped with an electrospray ionization (ESI) source. The ionization interface operated in negative polarity mode with the following settings: Scan parameters: full-scan mass range *m*/*z* 50–1500 in ultra-scan acquisition mode. Voltage settings: ion spray voltage set at 4.5 kV, declustering potential at 80 V. Collision-induced dissociation: fixed collision energy of 40 eV. Thermal conditions: ion source temperature maintained at 550 °C. The identification of the peaks was accomplished by comparing their retention times with those of individual reference compounds eluted in the mobile phase, and by introducing reference compounds into the samples for verification purposes.

### 2.3. Animal, Induction of Immunosuppression and Experimental Design

Fifty-five specific pathogen-free (SPF) male BALB/c mice, weighing 18–22 g and ten weeks of age, were procured from the Experimental Animal Center at Dalian Medical University, Dalian, China (SYXK [Liao] 2018-0002). All animals were housed at a temperature of 23 ± 1 °C and humidity of 50 ± 5%, under a 12 h light–dark cycle. They had ad libitum access to water and were fed a standard mouse chow diet. This experiment was approved by the Ethics Committee of Dalian Medical University (Ethics permit NO. AEE17013).

After a seven-day acclimatization period, the five groups were formed by a random division of the mice (*n* = 10): the CON group (normal control group) and CTX group (model group) received a daily intragastric administration of 200 µL of MRS medium, while the AS group (TCM group), PRO group (probiotic mixture, 1 × 10^9^ CFU/kg BW, *B. longum* SD5219/*L. fermentum* NCIMB5221/*L. paracasei* DM2806, strain ratio of 1/1/1), and PROAS group (probiotic-fermented *A. membranaceus* and Raphani Semen group) received an intragastric administration of equal volumes (200 µL) of *A. membranaceus* and Raphani Semen extracts, probiotics–MRS medium, or probiotic-fermented *A. membranaceus* and Raphani Semen broth once daily for 14 consecutive days from 7 days before CTX injection, respectively. At 10 a.m. of days 0, 1, and 2, mice in all groups (excluding the CON group), received an intraperitoneal injection of 100 mg/kg·bw CTX (Sigma-Aldrich, St. Louis, MO, USA) [21], while the CON group received an equivalent volume of saline via injection. The CTX solution was prepared as follows: CTX powder was dissolved in sterile normal saline to achieve a concentration of 10 g/L. The solution was then filtered through a 0.22 µm filter. Mice were administered the solution via intraperitoneal injection, based on their body weight, for three consecutive days. Throughout the experiment, the body weights (BWs) and daily activities of the animals were monitored daily. On days 3 and 7, mice were euthanized to collect serum, colon tissue, spleens, and feces for further analysis. Spleen index (mg/g) = spleen weight (mg)/body weight (g) × 10.

### 2.4. Histopathological Analysis of the Colon

The colon tissue was preserved using a 4% paraformaldehyde solution and then transformed into paraffin-embedded blocks. Subsequently, the sections underwent hematoxylin/eosin (H/E) staining to visualize any changes in morphology. Colonic histological damage was assessed based on previously established criteria [23].

### 2.5. Quantitative Real-Time Polymerase Chain Reaction (qPCR) Analysis

The RNeasy mini-Kit (Qiagen, Hilden, Germany) was utilized to extract total RNA from mouse colonic tissue. Subsequently, cDNA synthesis (Stratagene, La Jolla, CA, USA) was performed following the manufacturer’s instructions. For qPCR, Brilliant II SYBR Green qPCR Master Mix (Stratagene, CA, USA) was employed in an ABI StepOne Plus Sequence Detection System (Applied Biosystems, Foster, CA, USA). All samples underwent three rounds of amplification. β-actin was used as a housekeeping gene. The relative expression levels were determined using the 2^−∆∆Ct^ method [24]. The primers utilized in this study are detailed in Appendix A.

### 2.6. Flow Cytometry-Based Detection of Lymphocyte Subsets of Mice

Calycosin (purity ≥ 98.0%) and formononetin (purity ≥ 98.0%) were procured from Macklin Chemical Reagents (Shanghai, China). Spleen cells were harvested from five untreated mice and subsequently treated with a red blood cell lysis buffer (Solarbio, Solarbio Technology Co., Ltd., Beijing, China), counted, and processed to obtain the cell suspension. The cells were plated into 6-well cell culture plates at a density of 10^5^ cells/cm^2^. The concentrations of calycosin were 10, 50, and 100 µmol·mL^−1^ and formononetin were 10, 25, and 50 µmol·mL^−1^. A total of 1 × 10^6^ spleen cells·ml^−1^ were incubated in a 5% CO_2_ incubator at 37 °C for 24 h, after which they were incubated with different concentrations of the substances above for 24 h (or without added compounds in control experiments) [25].

Spleen cells from the other five groups of mice were processed identically, but no drugs were added to these samples. Subsequently, 1 × 10^6^ cells were analyzed by flow cytometry. Briefly, cells were incubated with an anti-mouse CD16/CD32 monoclonal antibody (MAb) to block Fcγ receptors, followed by staining on ice for 30 min using various combinations of MAbs. Subsequently, the cell suspensions were incubated with different antibodies, including PE-anti-mice CD19, FITC-anti-mice CD5, FITC-anti-mice CD138, FITC-anti-mice CD4, or PE-anti-mice CD8 (1:100 dilution for each antibody, eBiosciences, San Diego, CA, USA), on ice for 40 min without light [26]. Flow cytometry was performed using the FACS Calibur C6 Plus instrument (Becton Dickinson, Mountain View, CA, USA), and the data were analyzed using FlowJo software (version 10.9, Tree Star, Ashland, OR, USA).

### 2.7. Enzyme-Linked Immunosorbent Assay (ELISA)

The mouse peripheral blood was obtained by removing the eyeballs and then collected by centrifugation. The concentrations of lipopolysaccharide (LPS), interleukin (IL)-6, IL-17, IL-10, and tumor necrosis factor-α (TNF-α) were measured by using ELISA kits (provided by Jiancheng, Nanjing, China). In addition, phosphate-buffered saline (PBS) was used to thoroughly homogenize 100 mg of colon tissue; the resulting supernatant was collected after centrifugation to measure the levels of LPS, TNF-α, IL-6, IL-10, and IL-17. The production of IgM and IgG in the spleen cells after 24 h of co-culture with the calycosin and formononetin was detected. The detection procedure followed the instructions provided by the ELISA manufacturer. The absorbance was measured at 450 nm using an enzyme labeling instrument (Thermo Scientific™, Waltham, MA, USA).

### 2.8. Fecal DNA Extraction and Analysis of 16S rRNA Gene Sequence

The genomic DNA from the mouse feces of the five groups was extracted using the E.Z.N.A.^®^ Stool DNA Kit (Omega Bio-Tek, Norcross, GA, USA) following the guidelines provided by the manufacturer. After the identification of DNA isolations, the V3–V4 region in the 16S rDNA of gut bacteria was amplified and the amplicon was sequenced (Illumina HiSeq platform) at Novogene Bioinformatics Technology Co., Ltd. (Beijing, China). To generate Operational Taxonomic Units (OTUs) based on a 97% similarity threshold, we performed high-quality sequence clustering employing the UPARSE software (v7.0.1001). The analytical method employed in this study adhered to the previously established approach by Li et al. [22].

### 2.9. Detection of Short-Chain Fatty Acids (SCFA) from Fecal Samples

Solid feces samples (25 mg) of the five groups were supplemented with 1000 µL of a deproteinized solution containing 0.5% phosphoric acid and 10 µg/mL of 2-ethylbutyric acid. The supernatant was collected after centrifugation and the contents of acetic acid, propionic acid, butyric acid, isobutyric acid, valeric acid, and isovaleric acid, collectively classified as SCFA, were subjected to analysis using Agilent gas chromatography–mass spectrometry (GC-MS, 8890B-7000D) at Novogene Bioinformatics Technology Co., Ltd. following the methods established by Zhao et al. [27]. Analyte compounds were separated using an HP-FFAP capillary column (30 m × 0.25 mm × 0.25 µm). The GC column temperature was initially set to 80 °C and held constant, while the injection volume for each sample was 1 µL. The MS conditions involved an electron impact ionization source operating in positive mode, with the electron energy set at 70 eV. Standard curves for each SCFA (acetic acid, propionic acid, butyric acid, isobutyric acid, valeric acid, and isovaleric acid) were established using a series of calibration solutions, with concentrations ranging from 0.1 to 100 µg/mL. All analytes exhibited excellent linearity with correlation coefficients (R^2^) > 0.998. The limits of detection (LODs) and quantification (LOQs) were determined as signal-to-noise ratios of 3:1 and 10:1, respectively, with LODs ranging from 0.02 to 0.05 µg/mL and LOQs from 0.07 to 0.15 µg/mL across all target compounds. Recovery experiments were performed by spiking fecal samples with three concentration levels (low, medium, and high) of SCFA standards prior to extraction. The mean recovery rates for all analytes were 92.4–105.6%, with relative standard deviations (RSDs) < 5.2%, indicating satisfactory accuracy and precision.

### 2.10. Statistical Analysis

All data were expressed as the mean plus or minus the standard error of the mean (SEM), and a minimum of three separate experiments were conducted. Statistical significance was assessed utilizing Student’s *t*-test, one-way factorial analysis of variance (ANOVA), or nonparametric Mann–Whitney U tests, depending on whether the data followed a normal distribution. For multiple comparisons in ANOVA, the Bonferroni correction was applied to adjust the significance level and control the family-wise error rate. The significance level was set at *p* < 0.05 (after correction where applicable). All data analysis was conducted utilizing SPSS (version 17.0; SPSS Inc., Chicago, IL, USA).

## 3. Results

### 3.1. Change in the Composition of A. membranaceus and Raphani Semen After Probiotic Fermentation

A probiotic fermentation system was established for *A. membranaceus* and Raphani Semen, and three strains (*B. longum* SD5219, *L. fermentum* NCIMB5221, and *L. paracasei* DM2806) were used in this system. Their growth curves in MRS medium are shown in Figure 1A. As shown in Figure 1B–E, both the three strains and their mixture grew better in the *A. membranaceus* and Raphani Semen–MRS medium (AS) group than in the control medium (MRS) group, and the growth curves of the AS groups attained their maximum at the 24 h mark. This indicates that the bioactive compounds in the water extract of *A. membranaceus* and Raphani Semen were fully used by them, and probiotics were able to multiply more in AS. HPLC-MS was used to assess the differences in the primary constituents of AS and PROAS. The pH of PROAS was significantly lower than that of AS medium (*p* < 0.001, Figure 1F). Component parameters before and after fermentation are shown in Figure 1G–J. Figure 1G,I show that fermentation significantly increased the abundance of formononetin and calycosin in *A. membranaceus*; increased glucoraphenin and glucoraphanin; and decreased Z-glucosinolate, 3,4,5-trihydroxy-6-[(E)-C-[(E)-4-methylsulfoxyl-butyl-3-alkenyl]-n-sulfoxyl-carboamidyl]sulfamoxy-2-yl]methyl(E)-3-(4-hydroxy-3,5-dimethoxyphenyl)prop-2-alkenate, and 3,6′-sinaloyl sucrose in Raphani Semen (Figure 1H,J). These changes suggest that the probiotic mixture fermentation altered the utilization and conversion of TCMs.

### 3.2. PROAS Improves the Intestinal Barrier Function in Immunosuppressed Mice

The animal experimental design is shown in Figure 2A. The effects of PROAS on body weight changes are shown in Figure 2B,C; from day −7 to day 0, the weight of mice in all groups exhibited a slight decrease following the initiation of gavage. As the mice gradually adapted to the gavage procedure, their weight started to recover. CTX led to a pronounced diminution in body weight, an effect that was notably reversed upon administration of PROAS, thereby underscoring the efficacy of PROAS in mitigating CTX-induced weight loss (Figure 2C, *p* = 0.0019). Histopathological evaluation of the colon revealed that mice in the CON group displayed normal histological architecture, typified by intact and orderly arranged villi structures. In contrast, the CTX group exhibited pronounced intestinal mucosal damage, as indicated by the arrows in Figure 2D,F. This damage was characterized by a significant infiltration of inflammatory cells into both the mucosa and submucosa. (day 3: *p* < 0.0001; day 7: *p* = 0.0043, Figure 2D–G). The PROAS group demonstrated protection against crypt structure deterioration and histological inflammation in the colon (day 3: *p* = 0.0116). To further evaluate the impact of PROAS on intestinal barrier function, we determined the mRNA levels of tight junction (TJ)-related proteins. As shown in Figure 2H,I, the levels of mRNA expression of ZO-1 (*p* = 0.0062), claudin-1 (*p* < 0.0001), and claudin-4 (*p* = 0.0003) decreased on day 7 after CTX treatment, while those of ZO-1 (*p* = 0.0007), claudin-1 (*p* < 0.001), and claudin-4 (*p* < 0.001) were significantly increased by PROAS supplementation.

**Figure 1 biology-14-00312-f001:**
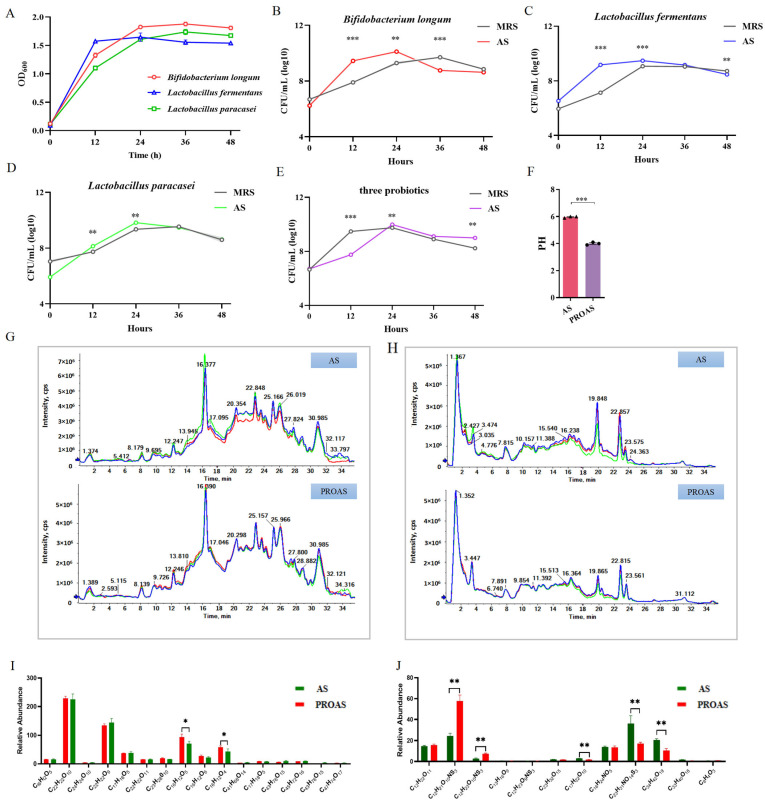
The analysis of chemical composition in *Astragalus membranaceus* and Raphani Semen fermented with probiotic strains. (**A**) The proliferation of three strains in the MRS medium; (**B**) the growth curves of *B. longum* SD5219; (**C**) the growth curves of *L. fermentum* NCIMB5221; (**D**) the growth curves of *L. paracasei* DM2806; (**E**) the growth curves of three probiotic mixtures; (**F**) the pH of *A. membranaceus* before (AS) and after (PROAS) fermentation with probiotic strains; (**G**) HPLC chromatograms for the analysis of the primary components of *A. membranaceus* before (AS) and after (PROAS) fermentation with probiotic strains, different color lines represent technical replicates of the same sample, with each sample analyzed in triplicate; (**H**) HPLC chromatograms for the analysis of the primary components of Raphani Semen prior to (AS) and following (PROAS) fermentation with three probiotic strains; (**I**) quantitative analysis of the principal components of *A. membranaceus* before (AS) and after (PROAS) fermentation with probiotic strains; and (**J**) quantitative analysis of the principal components of Raphani Semen prior to (AS) and following (PROAS) fermentation with three probiotic strains. All data are presented as mean ± SEM (*n* = 3). * *p* < 0.05, ** *p* < 0.01, *** *p* < 0.001; AS, *A. membranaceus* and Raphani Semen–MRS liquid medium; PROAS, probiotic-fermented *A. membranaceus* and Raphani Semen broth. *B. longum* SD5219, *L. fermentum* NCIMB5221, and *L. paracasei* DM2806.

### 3.3. PROAS Improves Immune Function in Immunosuppressed Mice

The spleen is the main immune organ, and the spleen index reflects the immune function of CTX-treated mice. As shown in Figure 3A,B and Figure 4A,B, the spleen index of mice treated with CTX after 3 (*p* < 0.001) and 7 days (*p* < 0.001) decreased significantly compared to that of the normal control, indicating that immunosuppression modeling was successful. The spleen index of the PROAS group was better compared to that of the CTX group (day 3: *p* = 0.0217; day 7: *p* = 0.0238). To further investigate whether PROAS affects the immune function of CTX-treated mice, the proportion of lymphocytes in mouse spleens was examined. The results revealed that the CTX group had a significantly lower percentage of CD19^+^ B lymphocytes compared to the CON group on days 3 (*p* = 0.0003) and day 7 (*p* = 0.0003) after CTX treatment. In addition, a significant increase in the proportion of CD19^+^ B lymphocytes was observed in the PROAS group (day 3, *p* = 0.0483; day 7, *p* = 0.0448). No noticeable alterations in the CD4^+^/CD8^+^ ratio among immunosuppressed mice across any of the five groups were observed (Figure 3C,D and Figure 4C,D).

As shown in Figure 3E,F and Figure 4E,F, further analysis of the immune parameters of CTX-treated mice revealed that the serum levels of interleukin (IL)-17 were significantly lower in the PRO and PROAS groups (day 3, *p* = 0.0421, *p* = 0.0317). Compared to the control group, a significant decrease in IL-10 (day 3: *p* = 0.0173, day 7: *p* = 0.0268) and TNF-α (day 3: *p* = 0.0007, day 7: *p* < 0.001) levels was observed in colonic tissue of the CTX group. Secretion of both proteins significantly increased with the continuous administration of PROAS. The levels of IL-6, IL-17, and LPS in the colon tissue were significantly reduced by PROAS supplementation, counteracting the increase caused by CTX (day 3, *p* < 0.001 and day 7, *p* < 0.001).

### 3.4. PROAS Modulates the Gut Microbiome Composition in Immunosuppressed Mice

To explore the effect of PROAS on the gut microbiota, fecal samples were collected on days 3 and 7 after CTX treatment and subjected to high-throughput sequencing targeting the 16S rDNA. The sequences were categorized into 1841 and 1175 OTUs on days 3 and 7, respectively. Rarefaction curves and abundance rank curves indicated that the depth of sequencing of these samples was sufficient to reflect diversity (Appendix A). The petal diagram indicated that there were 461 and 459 shared OTUs among the five groups, respectively (Figure 5A and Figure 6A). There was no significant difference in the α-diversity of the gut microbiota between the five groups (Figure 5B and Figure 6B). However, an increase in the Shannon index was observed in the PROAS group compared to the CTX group on day 7 after CTX treatment. The results of the principal coordinate analysis (PCoA) suggested no clustering of gut microbiota composition among the five groups (Figure 5C and Figure 6C). The changes in gut microbiome composition at the phylum level in mice treated with CTX after 3 and 7 days are shown in Figure 5D,E and Figure 6D,E, respectively. Compared to the PRO group, the PROAS group showed a reduction in the Firmicutes to Bacteroidetes (F/B) ratio. Notably, on day 7 (Figure 6E), the F/B ratio approached levels observed in the control group. To further elucidate the bacterial taxa contributing to this shift, we conducted an analysis of genus-level changes. When comparing the microbiota at the genus level, the abundance of *Alloprevotella* in the PROAS group exhibited an upward trend relative to the CTX group and other intervention groups; however, this increase was not statistically significant (Figure 5F,G and Figure 6F,G).

By comparing *A. membranaceus* and Raphani Semen before and after probiotic fermentation, it was observed that the abundance of butyrate-producing bacteria, including f-Muribaculaceae, f-Bacteroidaceae, f-Erysipelotrichaceae, g-bacteriodes, and f-Butyricicoccaceae, significantly increased at day 3. On day 7, the levels of bacteria that contribute to stabilizing intestinal epithelial homeostasis, such as s-Faecalibaculum rodentium, g-Faecalibaculum, and butyrate-producing bacteria o-Erysipelotrichales and f-Erysipelotrichaceae, were notably higher compared to pre-fermentation levels.

**Figure 3 biology-14-00312-f003:**
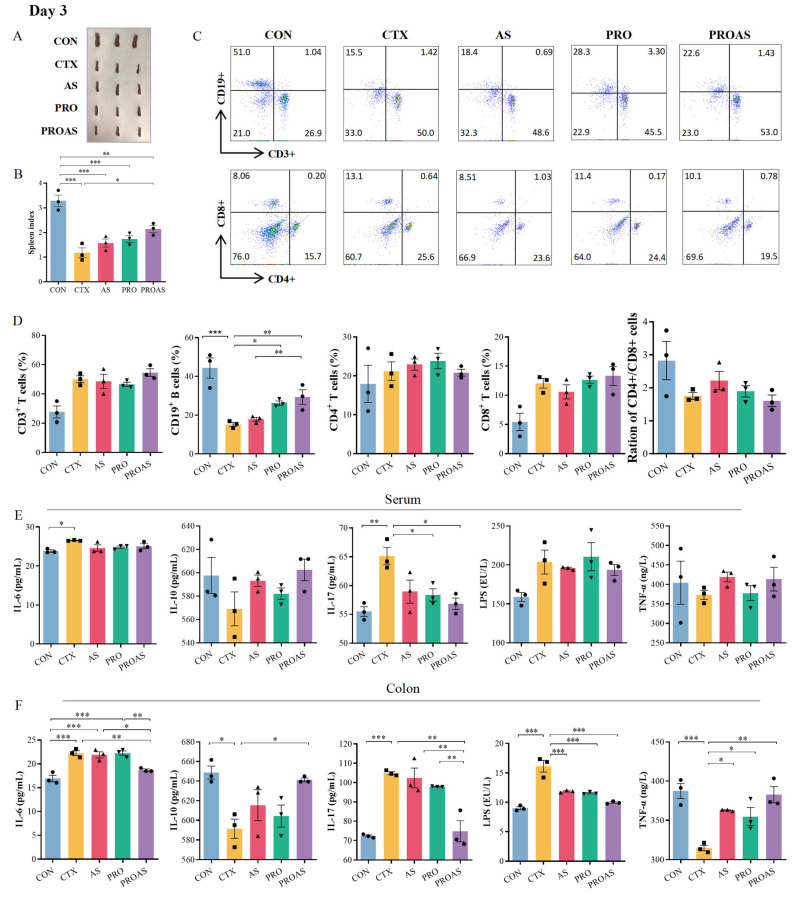
Immunomodulatory activity of PROAS in mice treated with CTX after 3 days. (**A**) Photograph of spleen; (**B**) spleen index; (**C**) representative flow cytometry plots, the color gradient indicates varying cell densities, with blue corresponding to low density and red representing high density. As the color shifts toward red, the cell density increases; (**D**) percentage of CD3^+^, CD19^+^, CD4^+^, and CD8^+^ cells, and the CD4^+^/CD8^+^ ratio in the spleen; (**E**) cytokines IL-6, IL-10, and IL-17, LPS, and TNF-α levels in serum; (**F**) colonic levels of cytokines IL-6, IL-10, and IL-17, LPS, and TNF-α, different symbol shapes are used to denote distinct samples within each group. Data are presented as mean ± SEM (*n* = 3), * *p* < 0.05, ** *p* < 0.01, *** *p* < 0.001.

**Figure 4 biology-14-00312-f004:**
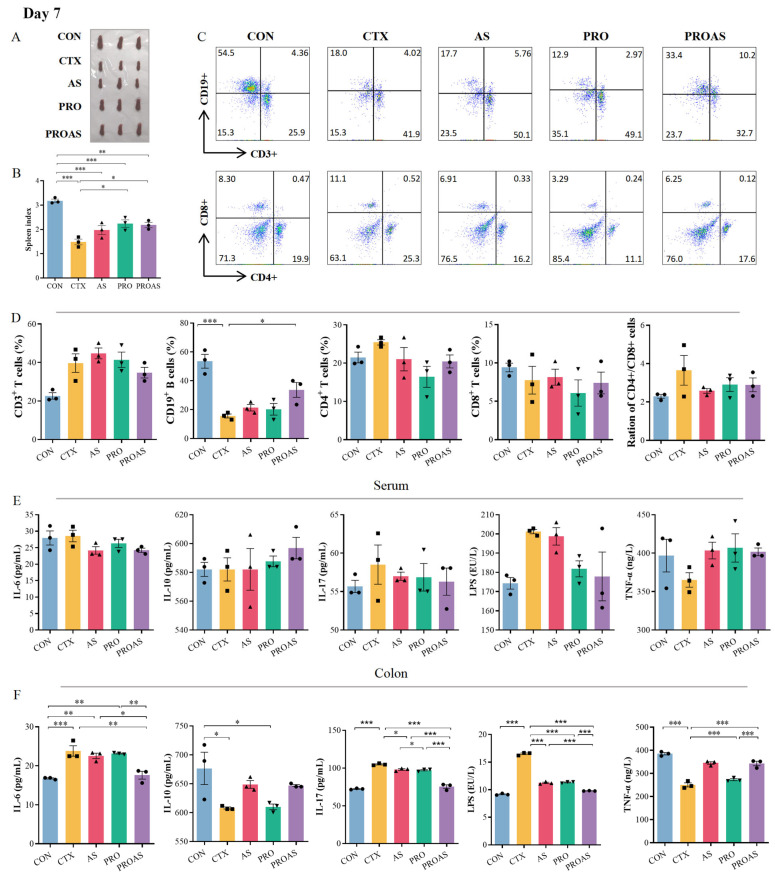
Immunomodulatory activity of PROAS in mice treated with CTX after 7 days. (**A**) Photograph of spleen; (**B**) spleen index; (**C**) representative flow cytometry plots, the color gradient indicates varying cell densities, with blue corresponding to low density and red representing high density. As the color shifts toward red, the cell density increases; (**D**) percentage of CD3^+^, CD19^+^, CD4^+^, and CD8^+^ cells, and the CD4^+^/CD8^+^ ratio in the spleen; (**E**) levels of cytokines IL-6, IL-10, and IL-17, LPS, and TNF-α in serum; (**F**) colonic levels of cytokines IL-6, IL-10, and IL-17, LPS, and TNF-α,different symbol shapes are used to denote distinct samples within each group. Data are presented as mean ± SEM (*n* = 3), * *p* < 0.05, ** *p* < 0.01, *** *p* < 0.001.

**Figure 5 biology-14-00312-f005:**
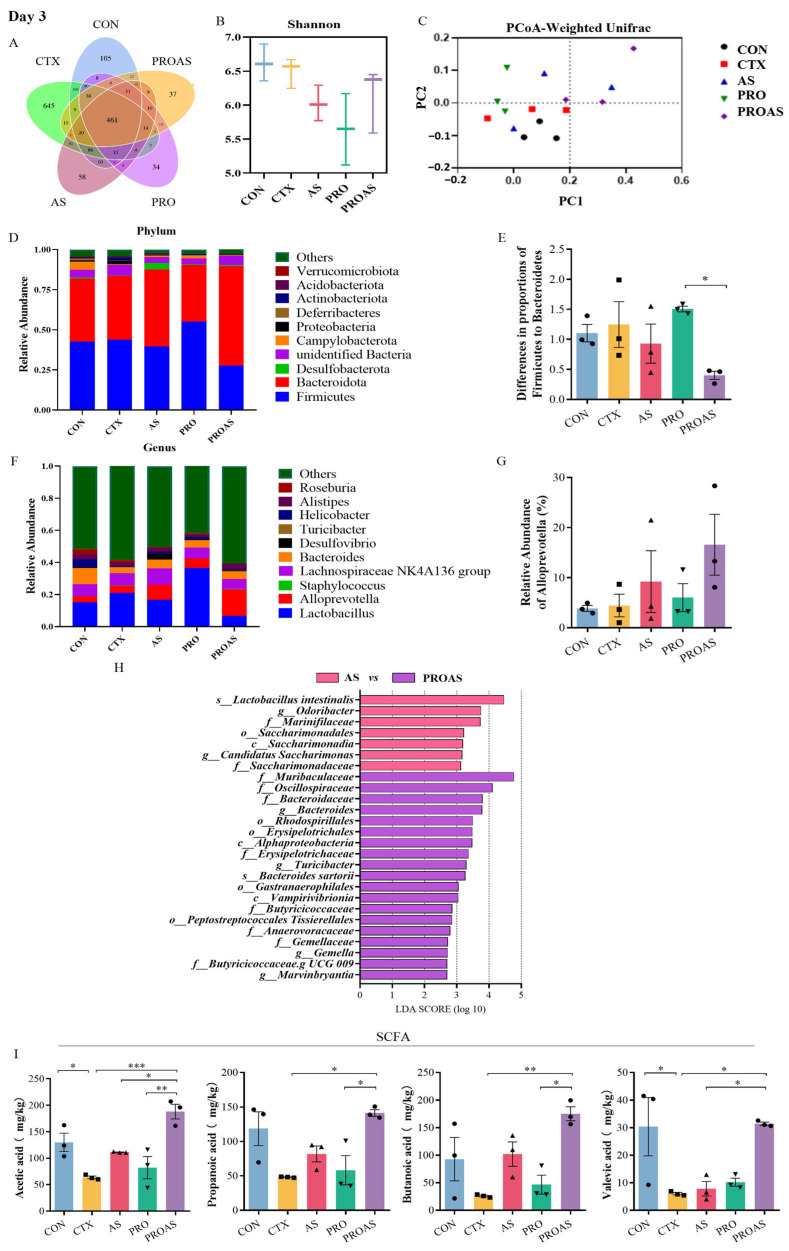
PROAS modulated the gut microbiota composition and intestinal SCFA production on day 3 after CTX treatment. (**A**) Petal diagram illustrating the distribution of shared and independent bacterial OTUs; (**B**) evaluation of α-diversity in the gut microbiota was performed using Shannon indices; (**C**) principal coordinate analysis (PCoA) based on weighted Unifrac distances to compare samples from different groups, the PCoA plot is divided into quadrants by dashed lines, with each quadrant representing distinct clusters of samples based on their similarity; (**D**) the average abundance of microbial communities at the phylum level; (**E**) the ratio of *Firmicutes* to *Bacteroidetes,* different symbol shapes are used to denote distinct samples within each group; (**F**) bar charts depicting the composition of the gut at the genus level; (**G**) relative abundance of *Alloprevotella* at the genus level, different symbol shapes are used to denote distinct samples within each group; (**H**) group-specific enriched taxa identified by a positive LDA score bar with different colors (LDA > 2.5); and (**I**) the exact amount of SCFA in cecal contents, including acetic, propionic, butanoic, and valeric acids, different symbol shapes are used to denote distinct samples within each group. Data are expressed as mean ± SEM (*n* = 3–7). * *p* <0.05, ** *p* < 0.01, *** *p* < 0.001.

**Figure 6 biology-14-00312-f006:**
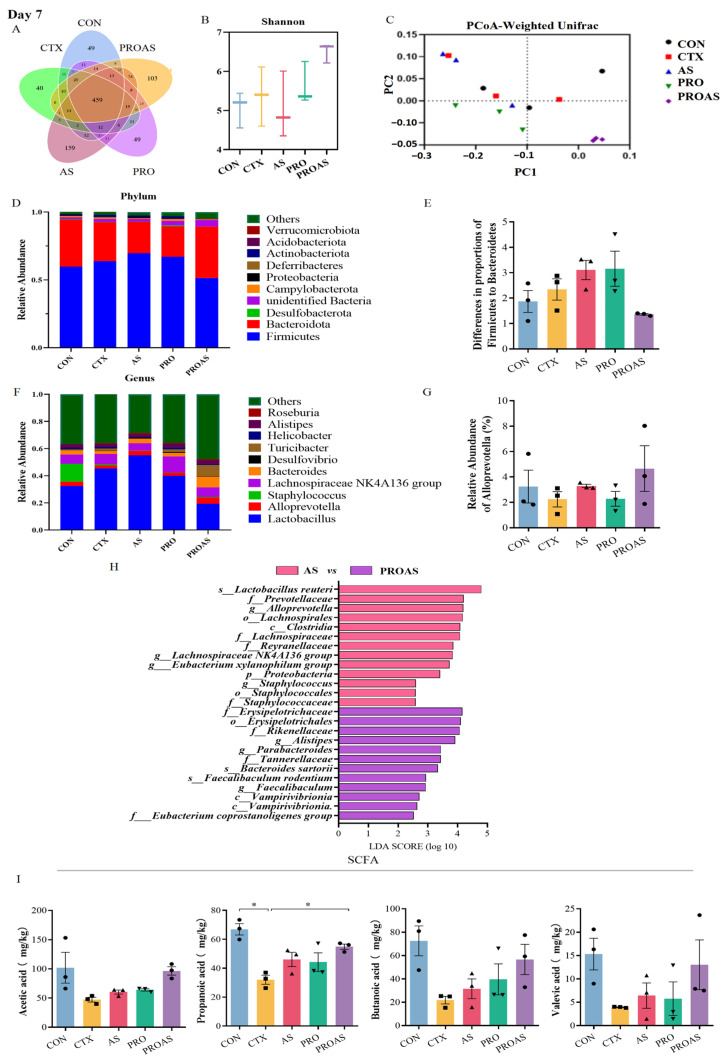
PROAS modulated the gut microbiota composition and intestinal SCFA production on day 7 following CTX treatment. (**A**) Petal diagram illustrating the distribution of shared and independent bacterial OTUs; (**B**) evaluation of α-diversity in the gut microbiota was performed using Shannon indices; (**C**) principal coordinate analysis (PCoA) was employed, utilizing weighted UniFrac distances, to compare microbial community compositions across different sample groups, the PCoA plot is divided into quadrants by dashed lines, with each quadrant representing distinct clusters of samples based on their similarity; (**D**) the average abundance of microorganisms at the phylum level; (**E**) the ratio of *Firmicutes* to *Bacteroidetes* in the gut microbiota; (**F**) bar charts depicting the composition of the gut at the genus level; (**G**) relative abundance of *Alloprevotella* at the genus level; (**H**) group-specific enriched taxa identified by a positive LDA score, represented by bars in different colors (LDA > 2.5); and (**I**) the concentrations of SCFA in cecal contents, including acetic, propanoic, butanoic, and valeric acids, were measured, different symbol shapes are used to denote distinct samples within each group. Data are presented as mean ± SEM (*n* = 3–7). * *p* < 0.05.

### 3.5. Effects of PROAS on Intestinal SCFA Production in Immunosuppressed Mice

SCFAs are essential bacterial metabolites in the intestine, primarily comprising acetic acid, propanoic acid, butanoic acid, isobutyric acid, valeric acid, isovaleric acid, hexanoic acid, and isohexanoic acid. As shown in Figure 5I and Figure 6I, we observed a significant reduction in the levels of acetic acid (*p* = 0.0411) and valeric acid (*p* = 0.0370) in the CTX-treated group compared to those in the CON group on day 3. Significant improvements were observed in acetic acid (*p* = 0.0006), propionic acid (*p* = 0.0118), butyric acid (*p* = 0.0056), and valeric acid (*p* = 0.0294) levels in the PROAS group. However, the suppression of SCFA by CTX was partially restored after 7 days of in vivo modeling. Nevertheless, the concentration of SCFA in the PROAS group remained elevated.

### 3.6. Effects of Calycosin and Formononetin from Astragalus on Mice Spleen Cells

After probiotic fermentation of *A. membranaceus*, the contents of compounds calycosin and formononetin were found to significantly increase. This may be attributed to the enhanced enrichment of these two active ingredients, which contribute to the immune function enhancement in mice. To investigate this further, mouse spleen cells were stimulated with different concentrations of compounds calycosin and formononetin. Immunophenotyping of mouse spleens was conducted via flow cytometry within the lymphocyte gate using fluorochrome-conjugated monoclonal antibodies specific for CD4, CD8, CD19, CD5, and CD138. Within the gated population, CD19^+^CD115^+^ cells were identified as B1 cells. For plasma cell identification, gated cells exhibited strong CD138 positivity with concomitant negative expression of CD19. Flow cytometry analysis revealed that three concentrations of calycosin could increase the proportion of T cells; each concentration of calycosin could increase the proportion of CD4^+^ T cells and CD8^+^ T cells. The ratio of CD4^+^/CD8^+^ increased significantly particularly at 50 and 100 µmol/mL (*p* < 0.001, *p* = 0.0016, Figure 7A), indicating an enhancement in cellular immunity. Formononetin at a concentration of 50 µmol/mL also significantly increased the proportion of T cells, although the CD4^+^/CD8^+^ ratio did not change significantly. In terms of humoral immunity, calycosin did not affect the proportion of B1 cells, whereas formononetin at 10 µmol/mL significantly increased the proportion of B1 cells and promoted IgM antibody production (Figure 7B). Additionally, calycosin at 10 µmol/mL significantly increased the proportion of plasma cells but did not stimulate significant IgG antibody production (*p* < 0.001). Nevertheless, formononetin at 10 µmol/mL significantly increased both the proportion of plasma cells and IgG antibody production (*p* < 0.001, *p* < 0.001, (Figure 7C)). To further elucidate the molecular mechanisms underlying these immunomodulatory effects, we analyzed the mRNA expression of key regulatory genes in stimulated spleen cells using qPCR. Formononetin treatment (10 µmol/mL) significantly upregulated the expression of *PRDM1* (Blimp-1), a master transcription factor driving plasma cell differentiation (*p* < 0.001 vs. control, Figure 7D). Conversely, calycosin (50 µmol/mL) markedly enhanced the expression of *H2-Ab* (a major MHC-II gene essential for antigen presentation to CD4^+^ T cells) (*p* < 0.001 vs. control, Figure 7D). These findings align with the observed increases in plasma cell proportions and IgG production (by formononetin) and T cell activation (by calycosin), suggesting that formononetin promotes humoral immunity via Blimp-1-dependent plasma cell maturation, while calycosin enhances cellular immunity through MHC-II-mediated T cell engagement.

## 4. Discussion

Probiotic fermentation technology was employed in this study to ferment *A. membranaceus* and Raphani Semen, aiming to enhance the production of beneficial microbial and plant-derived secondary metabolites. Consequently, we hypothesized that probiotic-fermented *A. membranaceus* and Raphani Semen (PROAS) would exhibit superior efficacy in alleviating CTX-induced immunosuppression compared to unfermented *A. membranaceus* and Raphani Semen (AS). As anticipated, PROAS not only enhanced immunostimulatory activity but also repaired intestinal barrier damage in CTX-induced immunosuppressed mice. Further investigation revealed that PROAS mitigated CTX-induced immunosuppression via SCFA-dependent or -independent regulation of intestinal immunity.

*A. membranaceus* and Raphani Semen are TCM prescriptions, used to treat the human body’s weakness of positive qi (deficiency) and pathogenic invasion (standard) at the same time. Fermentation promotes bacterial growth as well as the chemical conversion of TCM [22]. Our research results indicated that *A. membranaceus* and Raphani Semen significantly promoted the growth of *B. longum* SD5219, *L. fermentum* NCIMB5221, and *L. paracasei* DM2806 (Figure 1A–E)*. Bifidobacteria* and *Lactobacillus* are non-pathogenic, Gram-positive, and anaerobic bacteria that are predominantly utilized as probiotics owing to their beneficial effects on host health [28]. B. *longum*, L. *fermentum*, and L. *paracasei* have all been documented to exert protective effects on the intestinal mucosa and enhance the host immune responses [29,30,31].

A number of studies have shown that the fermentation of TCMs using *L. paracasei* and *L. fermentum* significantly improves their therapeutic effects. For instance, fermenting Curcuma longa with *L. paracasei* has been shown to ameliorate metabolic disorders in obese mice [32]. Likewise, the extract from Shenheling fermented with *L*. *fermentum* significantly improved its anti-obesity efficacy [33]. Therefore, it is essential to investigate the alterations in the primary constituents of *A. membranaceus* and Raphani Semen prior to (AS) and following (PROAS) co-fermentation by *B. longum* SD5219, *L. fermentum* NCIMB5221, and *L. paracasei* DM2806. In accordance with our prior research [20], the abundances of various predominant active ingredients in Raphani Semen, notably glucoraphenin and glucoraphanin, were significantly elevated in PROAS. Specifically, these glucosinolates are transported to the colon, where they undergo degradation by the intestinal microbiota. This process results in metabolites that can alleviate CTX-induced immunosuppression by enhancing the integrity of the intestinal barrier. The predominant flavonoid glucosides in *A. membranaceus* were biotransformed into their aglycone forms, specifically formononetin and calycosin, through enzymatic hydrolysis. This transformation likely results in higher absorption rates compared to the glycoside forms [22,34]. Formononetin has been extensively reported to possess anti-inflammatory and anti-tumor properties [35,36]. Calycosin has been reported to possess the capability to repair damage to the intestinal mucosa [37,38]. Probiotic fermentation elicited remarkable transformations in the chemical composition of Astragalus membranaceus and Raphani Semen, potentially augmenting the efficacy of PROAS in mitigating immunosuppression.

CTX is an alkylating agent with a wide range of efficacy against various diseases [39,40]. CTX kills tumor cells and also inhibits humoral and cellular immune responses, resulting in a decline in host immune function and compromised intestinal function, making it an excellent candidate for inducing immunosuppressive models in mice. Recently, much of the research on immune protection against CTX-induced immune toxicity has focused on Chinese herbs and their active ingredients as food materials and additives [11]. In this study, we employed a CTX-induced immunosuppressive mouse model to evaluate the efficacy and elucidate the underlying mechanisms of the immunostimulatory activity of PROAS. Probiotic fermentation techniques were used to cultivate *A. membranaceus* and Raphani Semen to produce advantageous microorganisms and secondary metabolites from TCM. Consequently, we hypothesized that PROAS would exert a more pronounced impact on mitigating CTX-induced immunosuppression. As anticipated, PROAS effectively repaired intestinal barrier damage and enhanced immunostimulatory activity in mice with CTX-induced immunosuppression. Remarkably, PROAS supplementation preserved the morphology and structure of the colon owing to an increase in the mRNA expression levels of TJ proteins, such as ZO-1, claudin-1, and claudin-4 (Figure 2D–H). Furthermore, as a potent immunosuppressive agent, CTX affects T and B cells and decreases immune responses by blocking DNA replication. Our findings demonstrate that PROAS plays an important role in alleviating immunosuppression by increasing the spleen index, increasing the growth of splenic B cells (Figure 3B,D and Figure 4B,D), and regulating the release of cytokines in the intestinal tract and serum. The mouse spleen functions as a secondary immune organ rich in lymphocytes and macrophages, and plays a crucial role in the activation of immune responses. Previous research has demonstrated the potential of CTX to decrease the spleen index in mice, whereas certain TCMs and their active components have shown promise for enhancing the immune organ index and preventing immune organ atrophy [41,42]. The detection of cytokines in serum and colonic tissues revealed that serum cytokine levels (i.e., in the peripheral circulation) did not exhibit significant variations among the groups. However, following different interventions, both pro-inflammatory and anti-inflammatory cytokine levels in colonic tissues underwent substantial changes, indicating that PROAS primarily manifests in intestinal immunity (Figure 3E,F and Figure 4E,F). This finding aligns with our previous research as well as that of several other scholars in the field [20,43,44].

Recently, scientific attention has focused on natural products that can regulate the composition of the gut microbiota. Wang et al. [45] reported the same therapeutic effects of *Bacillus subtilis*-fermented *A. membranaceus* in hyperuricemia via modulation of the gut microbiota. Chen et al. [41] found that polysaccharides from the roots of *Millettia speciosa* Champan ameliorated cyclophosphamide-induced intestinal injury and immunosuppression by modulating the gut microbiota and gut barrier function. The immune system relies on interactions with the gut microbiota to regulate immunity, enhance resistance to infections, and facilitate metabolism [46]. In our study, PROAS also mitigated CTX-induced gut microbiota dysbiosis by modulating the F/B ratio and enhancing the relative abundances of specific beneficial microbial taxa, in particular, bacteria that produce butyrate, for instance, *Alistipes*, Muribaculaceae, Bacteroidaceae, Erysipelotrichaceae, Butyricicoccaceae, *Faecalibaculum.* Correspondingly, we observed a significant increase in the concentration of SCFA following the intervention, with a particularly notable rise on day 3 (Figure 5 and Figure 6). *Alloprevotella* is a beneficial bacterial genus recognized for its production of SCFA and its positive association with IL-10, an anti-inflammatory cytokine. The increase in the *Alloprevotella* genus implies a corresponding rise in the abundance of SCFA-producing bacteria, which are known to confer health benefits (Figure 5G and Figure 6G) [47]. In this study, the increase in *Alloprevotella* relative abundance did not reach statistical significance, likely due to the limited sample size. Nonetheless, the observed upward trend may indicate a shift in the intestinal microbiota towards a healthier composition. A negative correlation between *Alistipes* levels and pro-inflammatory cytokines, such as TNF-α and IL-6, has been reported by Li et al. [48]. Furthermore, research has demonstrated that beneficial *Alistipes* bacteria provide protection against a range of diseases, including liver fibrosis, colitis, and cardiovascular diseases [49]. Muribaculaceae possess the remarkable ability to restore and fortify the intestinal mucosal barrier, break down complex polysaccharides, and promote the synthesis of beneficial SCFAs [50,51]. Bacteroidaceae are involved in polysaccharide degradation, SCFA production, the maintenance of intestinal barrier function, immune regulation, and alleviation of inflammation [52,53]. The Erysipelatoclostridiaceae family encompasses a diverse array of bacterial strains renowned for their exceptional capacity to synthesize butyric acid [54]. The Butyricicoccaceae family plays a crucial role in energy metabolism by providing essential SCFAs that serve as vital energy sources for the host’s intestinal epithelial cells. Additionally, this family exhibits anti-inflammatory properties, which can effectively mitigate intestinal inflammation [27,55]. These microorganisms are crucial in maintaining robust immune homeostasis and enhancing resistance against inflammatory disorders [56]. *Faecalibaculum* is a recently identified and formally named bacterial genus that belongs to the Erysipelotrichaceae family. In addition to promoting the production of SCFA, it also helps regulate immune response balance and inhibit excessive immune reactions, thereby mitigating inflammatory responses [57]. Therefore, we hypothesize that PROAS mitigated the microbiota imbalance caused by CTX and increased the abundance of beneficial bacteria while decreasing the levels of harmful bacteria, thereby restoring the microecological balance and preserving the health of the host. Importantly, these observations remain preliminary in nature. While our data suggest a potential role for these butyrate-producing bacteria in immune modulation via SCFA production, they underscore the need for future mechanistic studies. For example, fecal microbiota transplantation (FMT) or targeted antibiotic depletion models could directly validate the functional contributions of the beneficial bacteria, like *Alloprevotella*, *Faecalibaculum*, etc., to the observed immunomodulatory effects.

After fermenting *A. membranaceus* and Raphani Semen with probiotics, it was observed that the fermentation effectively alleviated the immunosuppression induced by CTX, particularly in restoring B cell function (Figure 3D and Figure 4D). To investigate whether this effect was due to an increase in bioactive compound production, we stimulated mice splenocytes with varying concentrations of formononetin and calycosin, both of which were significantly elevated following the fermentation of *A. membranaceus*. Our results demonstrated that these two substances, especially formononetin, markedly promoted B1 cell and plasma cell proliferation, enhanced IgM and IgG antibody expression, and enhanced overall humoral immunity (Figure 7B,C). Gu et al. conducted research and discovered that the active components in *A. membranaceus* include calycosin and formononetin. When administered to CT26 colon cancer-bearing mice, it was found to repair the intestinal mucosa, modulate the gut microbiota, promote the production of SCFA, and ultimately inhibit the growth and metastasis of colon cancer [38]. Formononetin has been reported to exhibit anti-inflammatory and antioxidant properties [58]. In this study, we observed that the levels of glucoraphenin and glucoraphanin in Rahpani Semen increased following fermentation. In our previous research, fermentation of Raphani Semen using different probiotics was shown to alleviate CTX-induced immunosuppression through regulation of the gut microbiota, production of SCFA, and repair of the mucosal barrier [20]. S. Giacoppo et al. reported that glucoraphanin can enhance both the expression and distribution of tight junction (TJ)-related proteins, including claudin-1, claudin-3, claudin-5, and ZO-1 [59]. Early-life administration of glucoraphenin-rich broccoli to mice can significantly prevent the onset of colitis and provide robust protection for the intestinal mucosa [60].

Our findings demonstrate that probiotic fermentation of Astragalus membranaceus and Raphani Semen (PROAS) effectively alleviates CTX-induced immunosuppression through multifaceted mechanisms. Notably, formononetin, a bioactive compound significantly enriched following fermentation, upregulated *PRDM1* (Blimp-1) expression in splenocytes (Figure 7D). Blimp-1 is a master transcriptional regulator of plasma cell differentiation, suppressing B cell identity genes such as PAX5 while activating immunoglobulin heavy-chain (IgH) transcription to drive IgG production [61]. This finding aligns with our observation that formononetin increased the proportion of plasma cells and elevated IgG levels (Figure 7C), thereby underscoring its role in enhancing humoral immunity. Conversely, calycosin, another flavonoid enriched by fermentation, elevated *H2-Ab* (MHC-II) expression in B cells (Figure 7D). MHC-II molecules on B cells facilitate antigen presentation to CD4^+^ T helper cells, which is a critical step for T cell-dependent immune activation [62]. This mechanism likely explains calycosin’s ability to amplify the CD4^+^/CD8^+^ T cell ratio (Figure 7A), thereby strengthening cellular immunity.

These molecular insights corroborate our earlier observations that PROAS restores intestinal barrier integrity and modulates gut microbiota. The observed increase in SCFA-producing bacteria (e.g., Butyricicoccaceae and Bacteroidaceae) and elevated SCFA levels (Figure 5I and Figure 6I) indicate that gut-derived metabolites synergize with the Blimp-1 and MHC-II pathways to enhance systemic immunity. Specifically, SCFAs, particularly butyrate, serve as an energy source for B cell proliferation [63], while MHC-II-mediated T cell activation primes adaptive responses against immunosuppressive insults. Collectively, PROAS orchestrates a tripartite immunomodulatory network: B cell-driven humoral immunity via Blimp-1-dependent plasma cell maturation and IgG secretion; T cell-mediated cellular immunity through MHC-II antigen presentation; and modulation of the gut–immune axis via SCFA and microbiota restoration. This integrative mechanism is consistent with previous research indicating that fermented TCMs enhance the bioavailability of bioactive compounds and increase probiotic activity [20,64].

## 5. Conclusions

In summary, the probiotic fermentation of *A. membranaceus* and Raphani Semen (PROAS) enhances immunomodulatory efficacy via three synergistic pathways: (1) Enrichment of bioactive compounds: Fermentation significantly increases the levels of formononetin, calycosin, glucoraphanin, and glucoraphanin. Formononetin promotes Blimp-1-dependent plasma cell differentiation, while calycosin facilitates MHC-II-mediated T cell activation. (2) Reprogramming of the gut microbiota: PROAS restores beneficial bacteria such as Butyricicoccaceae and elevates SCFA production, thereby providing metabolic support for immune cell function. (3) Repair of the intestinal barrier: The upregulation of tight junction proteins (ZO-1 and claudin-1/4) and anti-inflammatory cytokines (IL-10) mitigates CTX-induced mucosal damage. These findings underscore the potential of PROAS as a promising immunomodulatory agent, leveraging fermentation to enhance the bioactivity of TCM, promote microbiota crosstalk, and regulate molecular immune responses. Future studies should investigate its clinical translation, particularly in addressing chemotherapy-associated immunosuppression.

## Figures and Tables

**Figure 2 biology-14-00312-f002:**
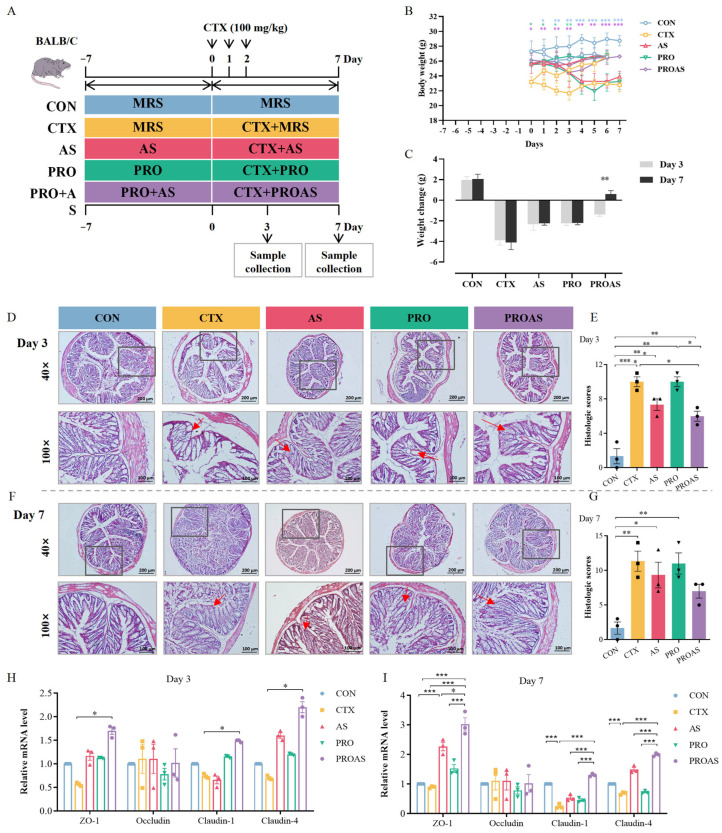
Improvement in intestinal barrier function by PROAS in immunosuppressed mice. (**A**) Experimental design; (**B**) body weights, where different colored asterisks denote statistically significant differences relative to the control group; (**C**) body weight change; (**D**) hematoxylin/eosin (H/E)-stained results of mouse colon on day 3 after CTX treatment, the boxes indicate specific areas requiring further observation, while the red arrows highlight locations where damage has been observed; (**E**) histopathological analysis of the H/E-stained sections on day 3 after CTX treatment; (**F**) H/E-stained results of mouse colon on day 7 after CTX treatment, the boxes indicate specific areas requiring further observation, while the red arrows highlight locations where damage has been observed; (**G**) histopathological analysis of the H/E-stained sections on day 7 after CTX treatment; (**H**) the relative mRNA expression levels of genes encoding ZO-1, occludin, claudin-1, and claudin-4 in colon tissues of mice on day 3 after CTX treatment, different symbol shapes are used to denote distinct samples within each group; and (**I**) the relative mRNA expression levels of genes encoding ZO-1, occludin, claudin-1, and claudin-4 in colon tissues of mice on day 7 after CTX treatment, different symbol shapes are used to denote distinct samples within each group. Data are expressed as mean ± SEM. * *p* < 0.05, ** *p* < 0.01, *** *p* < 0.001.

**Figure 7 biology-14-00312-f007:**
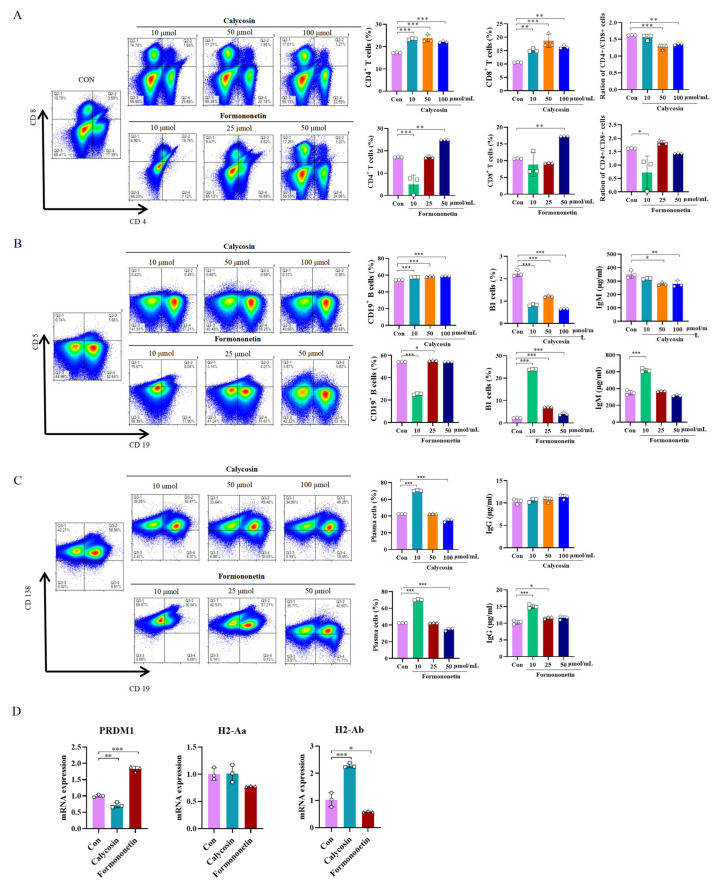
Immunomodulatory activity of the bioactive compounds generated through fermentation. (**A**) Representative flow cytometry plots, the color gradient indicates varying cell densities, with blue corresponding to low density and red representing high density. As the color shifts toward red, the cell density increases; percentage of CD4^+^, CD8^+^, and the CD4^+^/CD8^+^ ratio in the spleen. (**B**) Representative flow cytometry plots; percentage of CD19^+^, B1 cells, and levels of IgM. (**C**) Representative flow cytometry plots; percentage of plasma cells and levels of IgG. (**D**) Real-time qPCR analysis of *PRDM1* (Blimp-1) and MHC-II mRNA(*H2-Aa*, *H2-Ab*) levels, different symbol shapes are used to denote distinct samples within each group. Data are presented as mean ± SEM (*n* = 3), * *p* < 0.05, ** *p* < 0.01, *** *p* < 0.001.

## Data Availability

The original contributions presented in this study are included in the article/Appendix A; further inquiries can be directed to the corresponding authors.

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
