# Peer review of "Probiotic Fermentation of Astragalus membranaceus and Raphani Semen Ameliorates Cyclophosphamide-Induced Immunosuppression Through Intestinal Short-Chain Fatty Acid-Dependent or -Independent Regulation of B Cell Function"

_biology, 2025, doi:10.3390/biology14030312_

Round 1
Reviewer 1 Report
Comments and Suggestions for Authors
The article has merit, but there are a number of issues that need clarification.
1. Why is there a loss of body weight in the CTX group of mice, from day -7 to day 0, since no CTX was administered during this period (this follows from Figure2A?
2. In Part 2. Materials and Methods there is no part describing the methods for the results in Figures 1A-E, bacterial counting methodology. Тo description of where the culture of the strains used was obtained from, how the authors prepared the overnight culture for application.
3. I did not find a description of the methodology “Component analysis of the fermented A. membranaceus and Raphani Semen” in the article 24 Hiraku, A., Nakata, S., Murata, M., Xu, C., Mutoh, N., Arai, S., Odamaki, T., Iwabuchi, N.,Tanaka, M., Tsuno, T., Nakamura, M., 2023. Early probiotic supplementation of healthy term infants with bifidobacterium longum subsp. infantis M-63 is safe andleads to the development of bifidobacterium-predominant gut microbiota: a doubleblind, placebo-controlled trial. Nutrients 15 (6), 1402.
Line 212 "established by Zhao et al [36]." - The reference list does not include this author.
Line 203-204 The analytical method employed in this study adhered to the previously 203
established approach by Li et al. [24]. - The reference list does not include this author. Describe the methodology correctly.
Line 471 "Chang et al. [42] found... " - Uncorrect!
Line 494-498 "The Butyricicoccaceae family plays a crucial role in energy metabolism by providing essential SCFAs that serve as vital energy sources for the host's intestinal epithelial cells. Additionally, this family exhibits anti-inflammatory properties, which can effectively mitigate intestinal inflammation [51]. " Not correctly citing the article, there is no such statement in this article.
The article is very sloppily written, while it contains a lot of material, the approach described in the article is not new. The article completely repeats the article
Li Y, Lei Z, Guo Y, Liu Y, Guo X, Wang X, Che J, Yuan J, Wang C, Li M. Fermentation of Ganoderma lucidum and Raphani Semen with a probiotic mixture attenuates cyclophosphamide-induced immunosuppression through microbiota-dependent or -independent regulation of intestinal mucosal barrier and immune responses. Phytomedicine. 2023 Dec;121:155082. doi: 10.1016/j.phymed.2023.155082. Only Ganoderma lucidum and Astragalus membranaceus are changed.
The article is not finalized and requires careful attention to the literature cited. Also, the authors should describe the methods more correctly.
Author Response
Dear reviewer
I would like to express my deep gratitude to you for the constructive comments and suggestions.
A Point-by-Point Response to reviewers’ comments is given in the attachment, please see the attachment.
Thank you.

Reviewer 2 Report
Comments and Suggestions for Authors
In this study by Chen et al. (2025) the authors dedicated themselves to an extensive characterization of the potential of two herbs from traditional Chinese medicine fermented by different probiotic bacteria. The work is interesting and relevant, some points need to be better clarified before publishing the manuscript. See below:
Major comments
- lines 25-26, 87, 92, 228 and throughout the rest of the manuscript: whenever referring to a probiotic, please include the strain code/ID, as the probiotic action is strain-specific and not genus.
- Most methodology is not referenced, were they developed by this manuscript? Based on what literature was the model treatment carried out?
- general: was the pH of the extract neutralized after fermentation? how much was the initial pH and how much did it go to after fermentation?
Minor comments
- lines 63-64: please include references that support each of the claimed effects.
- Lines 75-76: please include references that support each of the claimed effects.
- lines 78 and wherever else necessary in the manuscript: replace the term microbe/germ with microorganism/microbiota or similar, otherwise it is inaccurate.
- line 83 and wherever else necessary: ​​use italics appropriately.
- line 104, 108: what is the filter pore size? What material was the filter made of?
- line 117: at what temperature?
- line 130: what is the initial age of the animals?
- line 146: what is the diluent for ctx?
- line 229 and wherever else necessary: ​​MRS
- Figures 1B-E: wouldn't it be more elegant to place the Y axis at log10 CFU/mL?!
- Figure 2D-F: Please include scale.
- Figure 2 and lines 262-265: please point out the observations made in the text in the figures, indicating the image with arrows/asterisk and complementing the caption.
- Figure 2B-C: no statistical differences are noted in the image, but in the text (lines 257-259) it says that there are differences, restoration, etc. Are they statistically significant?
Author Response
Dear reviewer
I would like to express my deep gratitude to you for the constructive comments and suggestions.
A Point-by-Point Response to reviewers’ comments is given in the attachment. Please see the attachment.
Thank you.

Reviewer 3 Report
Comments and Suggestions for Authors
1. when using cyclophosphamide (CTX) to induce immunosuppressive mouse model, the article did not fully explain the specific use method and dose of CTX. It is suggested that the author further describe the administration method and dose selection basis of CTX in detail in the method section.
2. the research design is highly similar to the previous work of the author's team (such as Ganoderma lucidum and radish fermentation [19]), so it is necessary to clarify the unique contribution of this study compared with previous studies.
3. the joint intervention control of "unfermented traditional Chinese medicine group (as) + probiotics group (pro)" was not set, and the synergistic effect of fermentation process and probiotics alone could not be clarified.
4. although the description of probiotic fermentation technology is detailed, it lacks specific information about strain source, fermentation time, temperature and other conditions. To ensure the reproducibility of the experiment, it is recommended to list these parameters in detail.
5. 16S rRNA sequencing only describes changes in the composition of the microbiota, and the functional contribution of specific microbiota (such as alloprevotella) is not verified by fecal bacteria transplantation (FMT) or antibiotic treatment models.
6. key parameters such as quantitative standard curve, detection limit and recovery rate of target components are not provided, which affects the reliability of data.
7. B cell function was assessed only by cd19+ ratio and igm/igg level, lacking detection of plasma cell differentiation (such as Blimp-1 expression), antibody affinity or antigen delivery related indicators (such as MHC-II).
8. in vitro experiments, only calycosin and formononetin were used to stimulate splenocytes, and it was not verified whether they function through the same mechanism in vivo, and the potential contribution of other fermentation products (such as polysaccharides) was not excluded.
9. the correction method for multiple comparisons (e.g. Bonferroni) is not stated. Some p values are close to the threshold (e.g. p=0.0483), and there may be a risk of false positives.
10. there is a contradiction between "no significant change in cd4+/cd8+ ratio" in Figure 3 and Figure 4 and "calycosin significantly increases cd4+/cd8+ ratio" in the discussion. It is necessary to check the consistency of data and description.
11. there is no direct evidence for the association between the change of f/b ratio and the "improvement of dysbacteriosis" (for example, the abundance of pathogenic bacteria was not detected), and the rise of alloprevotella did not reach statistical significance (the article mentioned "no significant difference"), so its biological significance should be carefully inferred.
12. the labels of some charts (such as figures 5 and 6) are fuzzy, and supplementary illustrations are required; Figure 7 flow results lack a representative description of gating strategies.
13. it is necessary to disclose the original sequencing data (such as NCBI accession number) and HPLC-MS peak identification details to enhance reproducibility.
Author Response

(The authors gave the same response as above.)

Round 2
Reviewer 1 Report
Comments and Suggestions for Authors
The authors have greatly improved the article.
Reviewer 2 Report
Comments and Suggestions for Authors
The authors answered all my questions satisfactorily.
Reviewer 3 Report
Comments and Suggestions for Authors
The author has already addressed the issue that concerns me